# Metagenomic Analysis for Diagnosis of Hemorrhagic Fever in Minas Gerais, Brazil

**DOI:** 10.3390/microorganisms12040769

**Published:** 2024-04-11

**Authors:** Felipe Campos de Melo Iani, Gabriel Montenegro de Campos, Talita Emile Ribeiro Adelino, Anielly Sarana da Silva, Simone Kashima, Luiz Carlos Junior Alcantara, Sandra Coccuzzo Sampaio, Marta Giovanetti, Maria Carolina Elias, Svetoslav Nanev Slavov

**Affiliations:** 1Laboratory of Virology, Ezequiel Dias Foundation (FUNED), Belo Horizonte 30510-010, MG, Brazil; felipeemrede@gmail.com (F.C.d.M.I.); talitaemile@yahoo.com.br (T.E.R.A.); anielly@usp.br (A.S.d.S.); 2Blood Center of Riberirão Preto, Faculty of Medicine of Ribeirão Preto, University of São Paulo, Ribeirão Preto 14051-140, SP, Brazil; gabrielmdecampos@usp.br (G.M.d.C.); skashima@hemocentro.fmrp.usp.br (S.K.); 3Instituto Rene Rachou, Fundação Oswaldo Cruz, Belo Horizonte 30190-002, MG, Brazil; alcantaraluiz42@gmail.com (L.C.J.A.); giovanetti.marta@gmail.com (M.G.); 4Climate Amplified Diseases and Epidemic (CLIMADE), Brasilia 70070-130, DF, Brazil; 5Laboratory of Cell Cycle (LECC), Center for Scientific Development (CDC), Butantan Institute, São Paulo 05585-000, SP, Brazil; sandra.coccuzzo@butantan.gov.br (S.C.S.); carolina.eliassabbaga@butantan.gov.br (M.C.E.); 6Department of Science and Technology for Humans and the Environment, University of Campus Bio-Medico di Roma, 00128 Rome, Italy; 7Laboratório de Arbovírus e Vírus Hemorrágicos, Instituto Oswaldo Cruz, Rio de Janeiro 21040-900, RJ, Brazil

**Keywords:** hemorrhagic fever, viral metagenomics, next-generation sequencing, viral diagnosis, Brazil

## Abstract

Viral hemorrhagic fever poses a significant public health challenge due to its severe clinical presentation and high mortality rate. The diagnostic process is hindered by similarity of symptoms across different diseases and the broad spectrum of pathogens that can cause hemorrhagic fever. In this study, we applied viral metagenomic analysis to 43 serum samples collected by the Public Health Laboratory (*Fundação Ezequiel Dias*, FUNED) in Minas Gerais State, Brazil, from patients diagnosed with hemorrhagic fever who had tested negative for the standard local hemorrhagic disease testing panel. This panel includes tests for Dengue virus (DENV) IgM, Zika virus IgM, Chikungunya virus IgM, yellow fever IgM, Hantavirus IgM, *Rickettsia rickettsii* IgM/IgG, and *Leptospira interrogans* IgM, in addition to respective molecular tests for these infectious agents. The samples were grouped into 18 pools according to geographic origin and analyzed through next-generation sequencing on the NextSeq 2000 platform. Bioinformatic analysis revealed a prevalent occurrence of commensal viruses across all pools, but, notably, a significant number of reads corresponding to the DENV serotype 2 were identified in one specific pool. Further verification via real-time PCR confirmed the presence of DENV-2 RNA in an index case involving an oncology patient with hemorrhagic fever who had initially tested negative for anti-DENV IgM antibodies, thereby excluding this sample from initial molecular testing. The complete DENV-2 genome isolated from this patient was taxonomically classified within the cosmopolitan genotype that was recently introduced into Brazil. These findings highlight the critical role of considering the patient’s clinical condition when deciding upon the most appropriate testing procedures. Additionally, this study showcases the potential of viral metagenomics in pinpointing the viral agents behind hemorrhagic diseases. Future research is needed to assess the practicality of incorporating metagenomics into standard viral diagnostic protocols.

## 1. Introduction

Viral hemorrhagic fever in Brazil poses a significant public health problem, with the most notable types including dengue hemorrhagic fever (dengue shock), yellow fever, and arenavirus hemorrhagic fever [1]. In this large territory that encompasses different biotopes, there is also circulation of a number of emerging viral agents that may represent a diagnostic challenge. The clinical manifestations of hemorrhagic fever may range from a mild febrile disease to life-threatening conditions characterized by coagulation abnormalities and hemodynamic instability that can lead to lethal outcomes [2]. Due to the similarity of the clinical manifestations, etiological diagnosis of arboviral fever presents significant challenges, and it is difficult to be established based only on clinical signs [3]. One of the most suitable methods for identifying novel, rare, or unsuspected pathogens in clinical samples is viral metagenomics. This unbiased method based on next-generation sequencing reveals all the genetic abundance of a given sample, including bacterial and viral sequences. As such, it has been used in clinical practice for etiological diagnosis of infectious diseases [4,5], showing high sensitivity and more detailed information for the identified pathogenic agents [4]. In this study, we aimed to apply viral metagenomics for investigating cases of suspected hemorrhagic fever without a confirmed diagnosis in the Brazilian state of Minas Gerais. The results obtained in this study can be useful in establishing differential diagnoses and detecting emerging or unsuspected viruses.

## 2. Materials and Methods

### 2.1. Clinical Samples

In this study, we performed metagenomic analysis on 43 serum samples from patients clinically suspected of having hemorrhagic fever, collected between February 2021 and April 2022. All samples were obtained from the most important geographic regions of the State of Minas Gerais with the help of the Fundação Ezequiel Dias, Belo Horizonte, Brazil (Figure 1). All samples were tested according to the institution’s established hemorrhagic fever protocol (Zika virus IgM, Chikungunya virus IgM, yellow fever IgM, Hantavirus IgM, *Rickettsia rickettsii* IgM and IgG, and *Leptospira interrogans* IgM and agglutination tests and the respective PCR for DENV, Zika, Chikungunya, yellow fever, *Rickettsia rickettsii*, and *Leptospira interrogans*). The detection of DENV IgM was conducted utilizing the Panbio™ Dengue IgM Capture ELISA kit (Abbott, Lake Bluff, IL, USA). For the identification of Zika virus IgM antibodies, the Zika ELISA kit (Vircell, Granada, Spain) was employed, whereas the detection of Chikungunya virus IgM antibodies was performed through the use of the Chikungunya ELISA kit (Euroimmun, São Paulo, Brazil). The molecular identification of these arboviruses was achieved using the Zika Dengue Chikungunya (ZDC) molecular detection kit (Bio-Manguinhos, Rio de Janeiro, Brazil). Serological detection of Hantavirus IgM antibodies was performed with the Hantavirus IgM detection kit (Abcam, Cambridge, UK), and the identification of anti-yellow fever virus IgM antibodies was carried out employing an in-house MAC-ELISA, adhering to the protocols recommended by the Centers for Disease Control and Prevention (CDC) in Atlanta. Molecular analysis for yellow fever virus was executed based on the primers and probes delineated by [6]. The serological identification of Rickettsia rickettsii was achieved via an in-house designed assay for IgM/IgG detection, and the molecular detection of rickettsial DNA was conducted using the method established by [7]. Lastly, the diagnosis of anti-Leptospira IgM was performed using the Panbio™ Leptospira IgM ELISA (Abbott, Lake Bluff, IL, USA), with the molecular diagnosis following in-house protocols as described in the literature by [8,9]. The samples are submitted to molecular testing if collected up to five days after symptoms onset. The results for all these samples were negative for the tested hemorrhagic fever pathogens. The samples comprised those from 30 male and 13 female patients, with a mean age of 45.79 years (ranging from 4 to 86 years, IQR (24–63) years). This study was approved by the Institutional Ethics Committee of the Federal University of Minas Gerais (process number CAAE 32912820.6.1001.5149).

### 2.2. Nucleic Acids Extraction, Amplification, and Next-Generation Sequencing

Initially, 600 μL of serum was pre-treated with Turbo DNAse (ThermoFisher Scientific) to remove free host and bacterial DNA. After DNAse inactivation, 2 or 3 individual samples were pooled into 18 pools based on the geographic localization of the samples to reduce sequencing cost. Nucleic acids were extracted from the total volume of the pools using the High Pure Viral Nucleic Acid Large Volume Kit (Roche, Basel, Switzerland) with minor modifications, such as the use of GenElute Linear Polyacrylamide carrier (LPA) (Merck, Darmstadt, Germany) for nucleic acids concentration and isopropanol for precipitation. After extraction, viral nucleic acids were recovered in nuclease-free water pre-heated to 70 °C in a final volume of 50 μL. Reverse transcription was conducted utilizing the Superscript III First-Strand Synthesis System (ThermoFisher Scientific, São Paulo, Brazil). The amplification of the cDNA was performed employing the QuantiTect Whole Transcriptome Kit (QIAGEN, Hilden, Germany), following an isothermal strategy. The sequence libraries were prepared using the Nextera XT DNA library preparation kit (Illumina, San Diego, CA, USA) and Illumina DNA/RNA UD indexes set B tagmentation (Illumina) following the manufacturer’s instructions. Sequencing of the dual-indexed libraries was conducted on the Illumina NextSeq 2000 sequencing platform using NextSeq 2000 P3 reagents (300 cycles, Illumina) according to the manufacturer’s instructions.

### 2.3. Bioinformatic Pipeline and Analysis

The raw sequence data underwent qualitative inspection using FastQC v.0.11.8 software [10]. Trimming, adapter removal, and sequence read filtering were performed with Fastp v.0.20.0 [11], retaining reads with a Phred quality score above 30, trimming of PolyX and PolyG tails, and performing base correction. Human reads were extracted using BWA 0.7.17-r1188 software [12] with the NCBI GRCh38.p14 human genome reference and the BWA-MEM algorithm [13]. Non-human (unmapped) reads were classified taxonomically and analyzed for virome content using Kraken2 v2.1.3 [14] against the RefSeq complete viral genomes/proteins database, last updated on 6th February 2024. Viral abundance analysis was conducted using the R programming language version 4.3.1—“Beagle Scouts” [15] and RStudio 2023.06.0+421 “Mountain Hydrangea” [16] as the integrated development environment (IDE). Key libraries used include readr v.2.1.4 [17], tidyverse v.2.0.0 [18], ggplot2 v.3.4.4 [19], and ggsci v.3.0.0 [20].

### 2.4. Viral Confirmation by Molecular Methods

After bioinformatic analysis, viruses of interest were confirmed by direct molecular methods. For Dengue virus (DENV) confirmation, we initially used real-time PCR with primers and probes that were conserved across all DENV serotypes [21]. Further, for molecular genotyping, we used primer and probe sets previously described in the literature [22]. The reaction conditions were as follows: 45 °C for 30 min, 95 °C for 5 min, and 40 cycles of 95 °C for 15 s and 60 °C for 1 min, performed on an Applied Biosystems 7500 cycler (ThermoFisher Scientific, São Paulo, Brazil).

### 2.5. Genomic Reconstruction

To determine the genotype in the confirmed DENV cases, we reconstructed their genomes, and those whose coverage was superior to 30× we submitted for phylogenetic analysis. DENV reads were extracted using KrakenTools [23] extract_kraken_reads command and aligned with the reference genome using ViralMSA [24]. The aligned reads were organized and subjected to variant calling using samtools v.1.12 [25]; a consensus sequence was generated using ivar v1.4.2 [26]. Bases not included in the consensus sequence were filled with Ns using MAFFT v7.520 [27]. Coverage and depth information was obtained using samtools [25].

### 2.6. Phylogenetic Analysis

For the phylogenetic analysis, we retrieved a total of 1305 complete DENV genomes meeting the following criteria: belonging to DENV serotype 2, devoid of ambiguities, obtained from South America, and covering the same time period as our study. Our search yielded genotype I (American), genotype II (Cosmopolitan), genotype III (Asian-American), and genotype IV (Asian), all originating from South America. The sequences were aligned using MAFFT v7.520 and were manually edited using AliView v.1.8 [28]. A maximum likelihood tree was reconstructed using iqtree2 v2.1.3 [29], with the substitution model automatically identified using ModelFinder [30]. Tree visualization was performed with ggtree [31].

## 3. Results

The next-generation sequencing performed on the 18 pools yielded sufficient data for a comprehensive viral metagenomic analysis. The origins of the samples, which were grouped into pools, are shown in Figure 1. A quantitative summary of the sequencing data obtained is presented in Table 1.

The virome of the tested patients was predominantly characterized by the presence of multiple commensal viruses, particularly from the *Anelloviridae* family: Genus Alphatorquevirus (Torque teno virus—TTV), Betatorquevirus (Torque teno mini virus—TTMV), and Gammatorquevirus (Torque teno midi virus—TTMDV). Additionally, high numbers of reads for human pegivirus (HPgV-1) were identified in pools 7 and 18, which is also considered a commensal virus (Figure 2). 

Among the viruses of clinical importance, DENV was identified with a high number of sequence reads in pool 15 (15,791,645 reads). This significant presence of DENV, notable as well for its clinical impact, prompted us to test all the individual patient samples from the positive DENV pool. Positive amplification for DENV RNA was observed in one sample. The index case showed positive amplification for DENV RNA with a cycle threshold of 20, and subsequent molecular DENV genotyping indicated positive amplification for DENV-2. Given that there was only one positive case in the pool, we were able to assemble and characterize the complete DENV genome. The mean depth of the assembled genome was 7009.8, with a depth of 1000X, accounting for 93.09% of the data. The coverage achieved was 99.08%, with only 0.02% of uncertain bases. The assembled DENV-2 genome measured 10,723 nucleotides in length. Phylogenetic analysis revealed that this sequence belongs to the DENV-2 cosmopolitan genotype, recently introduced in Brazil, clustering with sequences from Minas Gerais and Goias, aligning with the epidemiological scenario at the time (Figure 3). 

The DENV-positive patient, aged 71, was from the southern part of Minas Gerais state. Clinical symptoms emerged in mid-February 2022 (18 February 2022), and the sample was collected six days later. Tests for the hemorrhagic disease protocol (Dengue IgM, yellow fever IgM and IgG, *Rickettsia rickettsii* IgM and IgG, and Leptospira IgM) returned negative results. In this particular instance, the sample was not submitted to DENV molecular test due to protocol constraints. This was attributed to the following reasons: (1) The sample was collected six days after the symptom onset, while Public Health laboratory dictates that samples should be collected five days of symptom appearance for DENV RNA testing; (2) Additionally, the sample yielded negative result for DENV IgM, leading to its classification as negative. However, patient clinical condition was not deemed appropriate for DENV-RNA testing. The performed metagenomic analysis identified sequencing reads belonging to HIV and human mastadenovirus C; however, these agents are not currently associated with hemorrhagic diseases and are considered incidental findings. 

## 4. Discussion

In this study, we aimed to investigate the presence of viral agents involved in hemorrhagic disease through metagenomic analysis. Samples were collected from different locations in the state of Minas Gerais, Brazil, an area with extensive circulation of arboviruses, which complicates accurate clinical diagnosis. All the tested patients returned negative results for the locally established hemorrhagic disease panel, necessitating more robust analysis to explore the etiology of these severe conditions. The only viral agent identified capable of causing hemorrhagic disease was DENV, found in an elderly oncologic patient with hemorrhagic diathesis. DENV is a significant cause of hemorrhagic fever, potentially leading to dengue shock and multiple hemorrhages, with antibody-dependent enhancement following a second infection by a different serotype as a known cause [32]. This case presented with diagnostic peculiarities, notably the negative DENV IgM serologic panel and sample collection occurring more than 6 days post-symptom onset, which initially precluded molecular confirmation. However, metagenomics subsequently revealed a high number of sequence reads, which were further confirmed by real-time PCR, showing a low cycle threshold (Ct = 20). This low Ct value was indicative of acute DENV infection, suggesting that the patient’s oncologic condition might have delayed or prevented the production of anti-DENV antibodies, leading to a false negative initial serological result [33]. Therefore, the primary takeaway from this scenario is that the patient’s clinical condition should always be taken into consideration, and molecular investigation should be conducted regardless of the serological results.

Diagnosis of viral infections in patients with immunosuppression poses a formidable challenge, primarily due to the compromised serological responses often observed in this demographic. Consequently, it becomes critically important to integrate diagnostic methodologies endowed with the sensitivity to identify low viral loads, such as real-time PCR or digital PCR. Further augmenting the diagnostic arsenal, next-generation sequencing and metagenomic analysis offer unparalleled sensitivity and the comprehensive capability to detect a vast array of genomic sequences. This includes the genomes of viruses that carry significant clinical implications, thereby facilitating accurate identification of the infectious agents involved. Moreover, enhancement of human resource capabilities, particularly those personnel engaged in the execution of laboratory techniques and diagnostics, emerges as a pivotal factor. This enhancement, coupled with interdisciplinary integration of clinical and laboratory professionals, is essential for efficacious evaluation and accurate diagnosis of emergent viral pathogens in immunosuppressed populations. Such an integrated approach not only fosters precision in diagnostics but also significantly contributes to the broader objective of advancing patient care and management in the face of evolving viral threats.

The hemorrhagic outcome observed in this patient could be attributed to advanced age, which may exacerbate the clinical presentation [34]. Additionally, phylogenetic analysis identified the DENV-2 cosmopolitan genotype, which has been associated with more severe disease outcomes [35]. By 2022, this genotype was spreading in Brazil, with the first confirmed case reported in the state of Goias [36], highlighting the critical role of metagenomics in elucidating clinical cases when serological markers yield negative results. The data obtained in this case underscore the importance of metagenomics for elucidating clinical cases, particularly when the applied serological markers yield negative results. Consequently, viral metagenomics has been increasingly utilized for the discovery and characterization of emerging viruses. Furthermore, viral metagenomics has also proven useful for detection of established viruses, particularly respiratory ones [37,38].

As with all studies, ours also presents certain limitations. A primary limitation is the focus solely on viral pathogens, with steps primarily geared towards depletion of bacterial DNA. Hemorrhagic diseases can also be caused by various bacteria, including but not limited to rickettsial infections (such as Brazilian spotted fever), shigellosis, hemorrhagic salmonellosis, meningococcemia, and leptospirosis, all of which pose significant challenges in diagnosis. A more meticulous search in the bioinformatic data obtained for bacteria revealed abundant reads of Leptospira interrogans in pool 18, and pools 8 and 13 showed high sequence read numbers for Rickettsia rickettsii, which causes Brazilian spotted fever, a hemorrhagic disease associated with high mortality rates [39]. However, specific confirmation of these infections was beyond the scope of our study. Additionally, our investigation was constrained by the utilization of an established Reference Sequence Database containing known viral genomes. Consequently, unclassified reads were not thoroughly examined, potentially resulting in oversight of emerging viruses possibly associated with hemorrhagic diathesis. These limitations underscore the need for comprehensive approaches in future studies, which may involve expanding the scope to include bacterial pathogens and refining methodologies to capture emerging viruses. Such efforts are essential for advancing our understanding of the complex etiology of hemorrhagic fever and improving diagnostic capabilities.

## 5. Conclusions

In conclusion, viral metagenomics has demonstrated its value in identifying the viral origin of hemorrhagic fever in the State of Minas Gerais, Brazil, as evidenced by the confirmation of a DENV-2 cosmopolitan genotype in a patient with negative DENV IgM serology. Additionally, viral metagenomics provides deeper insights into the etiology of diseases that cannot be attributed to routine diagnostic tests alone. However, additional confirmatory tests, mostly direct molecular methods, are recommended to ensure accuracy and reliability. Moving forward, further research is necessary to effectively incorporate viral metagenomics into routine clinical diagnostics. This includes refining protocols, optimizing bioinformatics pipelines, and establishing standardized guidelines for result interpretation. By doing so, we can harness the full potential of viral metagenomics to enhance our understanding of infectious diseases and improve patient care outcomes.

## Figures and Tables

**Figure 1 microorganisms-12-00769-f001:**
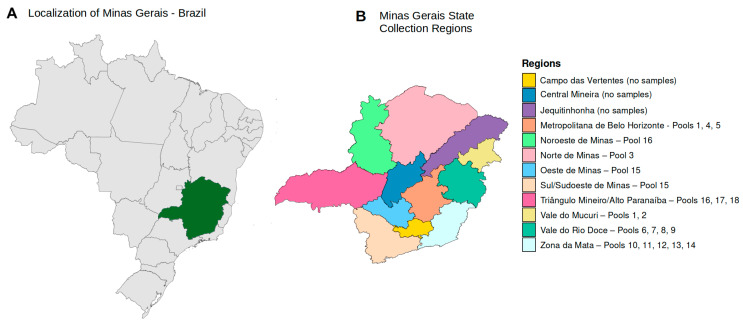
Spatial location of Minas Gerais State and the collection municipalities. (**A**) Location of Minas Gerais State on the background of the Brazilian national territory. (**B**) The State of Minas Gerais with its subdivisions. On the right the legend shows the number of pools linked to each geographical region. No samples were obtained from Campo das Vertentes, the Central region and Jequitinhonha. The state capital of Minas Gerais, city of Belo Horizonte, is in the Central Region (dark blue color).

**Figure 2 microorganisms-12-00769-f002:**
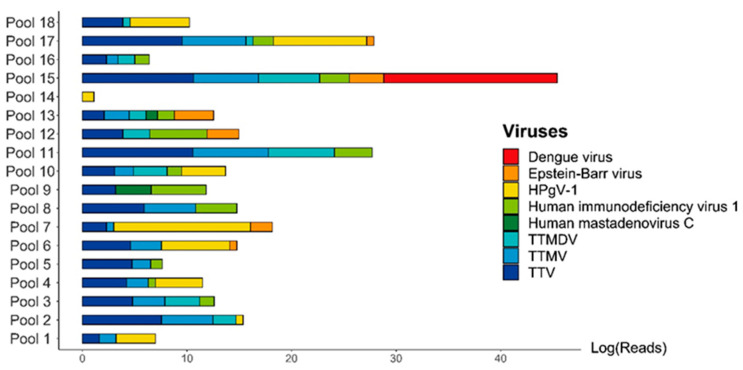
Horizontal bar plot of the abundance of the normalized virome of viruses that were identified among patients with clinical diagnosis of hemorrhagic fever from the state of Minas Gerais. It is possible to observe the extraordinary abundance of DENV in pool N 15. Additionally, we identified many commensal viruses of the anellovirus group, like torque teno virus and torque teno midi and mini viruses (TTV, TTMDV, and TTMV). X axis: natural log of the read number for each identified virus. Y-axis: the respective pool numbers.

**Figure 3 microorganisms-12-00769-f003:**
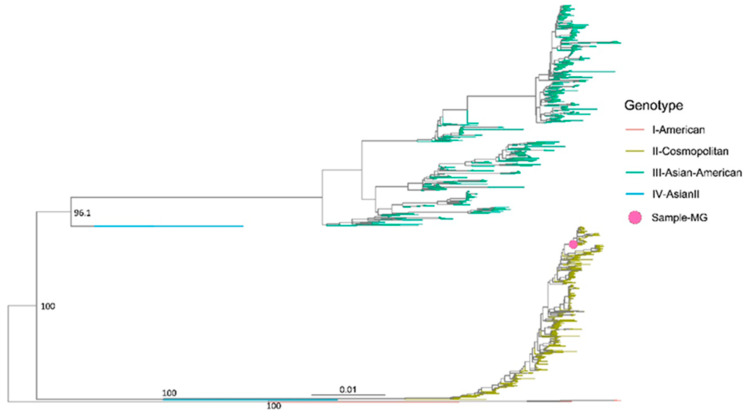
Maximum-likelihood tree of Dengue virus serotype 2. The analyzed DENV-2 complete genome was classified as Dengue-2 cosmopolitan genotype and clustered along with sequences of this genotype obtained from the Brazilian States of Minas Gerais and Goias.

**Table 1 microorganisms-12-00769-t001:** Quantity of viral reads obtained after the quality control and mapping of the tested pools.

Pool Number	Total Number of Reads (Millions)	Number of Reads after Filtering and Trimming (Millions)	Unmapped Reads (Millions)	Viral Reads
1	103.010	84.575	0.183	56 (0.03%)
2	121.620	104.490	1.440	8275 (0.57%)
3	130.640	117.370	15.600	602 (0.003%)
4	129.870	107.480	1.070	15,353 (1.44%)
5	116.220	94.794	1.270	25,457 (2.00%)
6	167.540	143.230	1.160	892 (0.08%)
7	176.370	154.430	1.720	485,921 (28.18%)
8	103.140	90.939	21.100	20,548 (0.10%)
9	129.920	113.770	9.105	758 (0.01%)
10	119.890	101.820	3.400	354 (0.01%)
11	113.000	95.453	1.960	46,446 (2.37%)
12	124.890	104.510	1.240	5930 (0.48%)
13	129.450	109.390	1.060	1130 (0.11%)
14	146.160	120.320	0.813	128 (0.02%)
15	114.320	97.523	16.600	15,844,322 (95.72%)
16	51.271	44.317	0.731	96 (0.01%)
17	171.710	143.690	2.340	22,799 (0.97%)
18	126.580	100.410	0.925	7782 (0.84%)

## Data Availability

The genome of DENV-2 was deposited in the Global Initiative on Sharing All Influenza Data (GISAID) repository, accessible at https://gisaid.org/, accessed on 7 March 2024 with the identifier EPI_ISL_18984971. Additionally, it has been deposited in the National Center for Biotechnology Information (NCBI) repository with the accession number PP546320.

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
