# Peer review of "Metagenomic Analysis for Diagnosis of Hemorrhagic Fever in Minas Gerais, Brazil"

_microorganisms, 2024, doi:10.3390/microorganisms12040769_

Round 1

Reviewer 1 Report

Comments and Suggestions for Authors

Summary:

This paper presents an unbiased sequencing analysis on a panel of patient samples with clinical signs of hemorrhagic fever, who initially tested negative for typical suspects of endemic viruses and bacteria known to cause hemorrhagic fever. From 43 serum samples, the authors identified one patient to have DENV-2. The DENV-2 genome was assembled, and phylogenetic analysis show the sequence to group with DENV-2 virus sequences from the same region of Brazil at that time. Overall, it is a brief study that could be improved with adding more details of the diagnostic assays, patient clinical disease (were all hospitalized patients?) and results of the identified sequence.

Include DENV-2 in the abstract describing the results. 

Methods:

Prior to the NGS, patients were tested by both molecular PCR and serology IgM assays for DENV, ZIKV, CHIKV, YFV, “Hantavirus”, Rickettsia and Leptospira. Please include a description of the assays (PCR primer sequence) or citation referring to these assays. What strains are included, especially for hantaviruses and DENV?

The DENV molecular assay should be described since it failed to detect the sequence in this paper.  

Was the host ribosomal RNA depleted?

Overall, the details regarding the NGS library and bioinformatic analysis looks compete.

However, a significant result is lacking.

The DENV-2 sequence found in the pool and used in the phylogenetic analysis: How complete was the sequence to the reference genome and what was the coverage?

Where did it differ from primers and probed used in the molecular diagnostic assays?

Was Sanger sequencing used to compete the sequence or 3’ and 5’ ends of the genome? Primers used need to be included (section 2.3)

Line 117-118. Replace molecular serotyping with genotyping.

The reads from NGS need to be uploaded to the public database of NCBI Bioproject and the DENV sequence needs to be submitted to GenBank, in addition to the one sequence submitted to GISAID.

In the phylogenetic analysis, methods 2.6 says 1305 DENV genomes were retrieved originating from South America, but the tree contains sequences from Asia and Americas. Please correct and include the accession identifiers of the sequences used in the phylogenetic analysis somewhere.

The detection of Anelloviruses in most pools is interesting and may reflect the clinical status of these acutely symptomatic patients and recently described as a marker of immune incompetence. As a weakened immune system fails to control the virome normally residing in the body, these virus levels have been found to increase. Have the authors found this in other studies or could they relate the detection of TTV to a clinical relevance of the patients in the study?

However, Anelloviruses are circular DNA viruses. The authors need to discuss what reads were detected from TTV- how were they identified in a NGS library that amplified only polyA- RNAs. Did they detect TTV mRNA transcripts?

Table 1. Please reduce the number of decimal places.

Discussion:

The authors need to address why they failed to detect DENV-2 in the patient using the initial panel of diagnostic assays, both by their IgM serological assay as well as the molecular (RT-PCR?) assay. Did the new sequence contain SNPs at significant primer binding sites? Did the patient fail a serology control given their supposed immune suppression by being an oncology patient?  More explanation is needed.

The library method chosen for the metagenomic study was designed for gene expression analysis of human polyA mRNA transcripts. This poses a great limitation since many pathogenic viruses do not encode a polyA mRNAs and would fail to be detected. The upstream use of DNAse is mentioned. What other endemic viruses were missed if they had chosen a different strategy?

What do the authors plan to do with the large percentage of non-human, unmapped reads? What other approach could help identify the etiological agent of these patients?

In a clinical diagnostic lab, what improvements are needed to detect highly pathogenic infectious agents in immune compromised patients? ...Especially if they could spread disease in a hospital setting.

Overall, this is an unsurprising result that needs a few more details for completeness. Often metagenomics studies fail to detect a pathogenic agent, but the data needs to be submitted to public, accessible databases to allow for reanalysis by different bioinformatic tools.  

Comments on the Quality of English Language

Line 117-118. Replace molecular serotyping with genotyping.

Line 87: typo: into

Author Response

REVIEWER#1

SUMMARY:

This paper presents an unbiased sequencing analysis on a panel of patient samples with clinical signs of hemorrhagic fever, who initially tested negative for typical suspects of endemic viruses and bacteria known to cause hemorrhagic fever. From 43 serum samples, the authors identified one patient to have DENV-2. The DENV-2 genome was assembled, and phylogenetic analysis show the sequence to group with DENV-2 virus sequences from the same region of Brazil at that time. Overall, it is a brief study that could be improved with adding more details of the diagnostic assays, patient clinical disease (were all hospitalized patients?) and results of the identified sequence.

We are sincerely grateful to reviewer #1 for the positive feedback provided on the manuscript. Thank you very much!

Include DENV-2 in the abstract describing the results.

We appreciate the suggestion made by the reviewer. DENV-2 has been included in the "Abstract" of the article (Lines 33-35, Page 1).

“…Bioinformatic analysis revealed a prevalent occurrence of commensal viruses across all pools, but notably, a significant number of reads corresponding to DENV serotype 2 were identified in one specific pool. …”

METHODS:

Prior to the NGS, patients were tested by both molecular PCR and serology IgM assays for DENV, ZIKV, CHIKV, YFV, “Hantavirus”, Rickettsia and Leptospira. Please include a description of the assays (PCR primer sequence) or citation referring to these assays. What strains are included, especially for hantaviruses and DENV?

We gratefully acknowledge the comment provided by the reviewer. In response to the feedback, we have clarified the selection and utilization of detection kits and algorithms within the applied "Hemorrhagic Fever" protocol. These details have been thoroughly elaborated upon in the manuscript, specifically within the "Materials and Methods" section. To further address the reviewer's suggestions, we have appended the following explanation to enhance understanding of the methodological choices and procedural intricacies.

Lines 77-92, Page 2: “…The detection of DENV IgM was conducted utilizing the Panbio™ Dengue IgM Cap-ture ELISA kit (Abbott). For the identification of Zika virus IgM antibodies, the Zika ELISA kit (Vircell) was employed, whereas the detection of Chikungunya virus IgM antibodies was performed through the use of the Chikungunya ELISA kit (Euroim-mun). The molecular identification of these arboviruses was achieved using the Zika Dengue Chikungunya (ZDC) molecular detection kit (Bio-Manguinhos). Serological detection of Hantavirus IgM antibodies was performed with the Hantavirus IgM de-tection kit (Abcam), and the identification of anti-Yellow fever virus IgM antibodies was carried out employing an in-house MAC-ELISA, adhering to the protocols rec-ommended by the Centers for Disease Control and Prevention (CDC) in Atlanta. Mo-lecular analysis for Yellow Fever virus was executed based on the primers and probes delineated by [6]. The serological identification of Rickettsia rickettsii was achieved via an in-house designed assay for IgM/IgG detection, and the molecular detection of rick-ettsial DNA was conducted using the method established by [7]. Lastly, the diagnosis of anti-Leptospira IgM was performed using the Panbio™ Leptospira IgM ELISA (Ab-bott), with the molecular diagnosis following in-house protocols as described in the lit-erature by [8,9]. …”

The DENV molecular assay should be described since it failed to detect the sequence in this paper.

We appreciate the valuable comment provided. The DENV molecular assay utilized in the Public Health laboratory does indeed detect Dengue and with high sensitivity (ZDC kit provided by the Brazilian Ministry of Health). However, in this particular instance, the sample was not subjected to the DENV molecular test due to protocol constraints. This is attributed to the following reasons: (1) The sample was collected 6 days after the onset of symptoms, while the Public Health Laboratory protocol dictates that samples should be collected within 5 days of symptom appearance for DENV RNA testing.

(2) Additionally, the sample yielded a negative result for DENV IgM, leading to its classification as negative. However, the patient's clinical condition was not deemed appropriate for DENV-RNA testing.

This study serves as an alert that patient condition should always be taken into consideration beyond established protocols. It is possible that, in this case, the patient was unable to produce anti-DENV IgM within the given timeframe and continued to harbor a high viral load. We apologize for any misunderstanding regarding the DENV testing algorithm by the reviewer and have incorporated this clarification into the revised version of the manuscript.

The following modifications were performed in the revised version of the manuscript

Lines 208-213, Page 7: “... “...In this particular instance, the sample was not submitted to DENV molecular test due to protocol constraints. This was attributed to the following reasons: (1) The sample was collected six days after the symptom onset, while Public Health laboratory dictates that samples should be collected five days of symptom appearance for DENV RNA testing; (2) Additionally, the sample yielded negative result for DENV-IgM, leading to its classification as negative. However, patient clinical condition was not deemed ap-propriate for DENV-RNA testing.   …”

Was the host ribosomal RNA depleted?

In this study, we conducted depletion of both host and bacterial DNA by applying a high concentration of DNAse directly to the clinical sample. It's important to note that we did not deplete specifically ribosomal RNA.

Overall, the details regarding the NGS library and bioinformatic analysis looks compete.

However, a significant result is lacking.

The DENV-2 sequence found in the pool and used in the phylogenetic analysis: How complete was the sequence to the reference genome and what was the coverage?

We identified one pool that exhibited a remarkably high number of DENV-2 sequence reads. This allowed us to assemble the complete DENV-2 genome, which was subsequently classified as belonging to the DENV-2 cosmopolitan genotype (Genotype II) based on the phylogenetic analysis conducted. The mean depth of the assembled genome was 7009.8, with a depth of 1000x accounting for 93.09% of the data. The coverage achieved was 99.98%, with only 0.02% of uncertain bases. These findings were also detailed in the “Results” section of the manuscript.

Lines 191-193, Page 6: “... The mean depth of the assembled genome was 7009.8, with a depth of 1000X account-ing for 93.09% of the data. The coverage achieved was 99.08%, with only 0.02% of un-certain bases. The assembled DENV-2 genome measured 10723 nucleotides in length. …”

Where did it differ from primers and probed used in the molecular diagnostic assays?

As previously explained, molecular testing for DENV was not conducted on the respective sample at the Public Health Laboratory. In fact, two completely different molecular systems were utilized for molecular confirmation and DENV genotyping, as described in detail in the "Materials and Methods" section. 

Was Sanger sequencing used to compete the sequence or 3’ and 5’ ends of the genome? Primers used need to be included (section 2.3).

The DENV-2 genome was assembled solely based on the sequence reads obtained from Next-Generation sequencing. Specific primers were not utilized to amplify the 3´ and 5´ extremities, and Sanger sequencing was not employed to complete the missing nucleotides.

The assembled DENV-2 genome measures 10723 nucleotides in length. The sequence terminates with a long non-coding RNA known as sfRNA1, spanning from nucleotide position 10,282 to 10,702. We added the following sentence in the “Results” section:

Lines 193-194, Page 6: “... The assembled DENV-2 genome measured 10723 nucleotides in length.…”

Line 117-118. Replace molecular serotyping with genotyping.

 We are grateful for the reviewer suggestion. “Molecular serotyping” was replaced by “Molecular genotyping” (Lines 139-140, Page 4).

The reads from NGS need to be uploaded to the public database of NCBI Bioproject and the DENV sequence needs to be submitted to GenBank, in addition to the one sequence submitted to GISAID.

We submitted the obtained DENV genome additionally to NCBI Genbank.  The access number is PP546320. This was added also to the revised version of the manuscript.

Lines 306-309, Page 9: “… The genome of DENV-2 was deposited in the Global Initiative on Sharing All Influenza Data (GISAID) repository, accessible at https://gisaid.org/ with the identifier EPI_ISL_18984971. Addi-tionally, it has been deposited in the National Center for Biotechnology Information (NCBI) repos-itory with the accession number PP546320. …”

In the phylogenetic analysis, methods 2.6 says 1305 DENV genomes were retrieved originating from South America, but the tree contains sequences from Asia and Americas. Please correct and include the accession identifiers of the sequences used in the phylogenetic analysis somewhere.

We are grateful for the valuable feedback from the reviewer. In response to this, we conducted a thorough search on GISAID EpiArbo for all DENV-2 genotypes present in South America. Our search yielded the genotypes I (American), genotype II (Cosmopolitan), genotype III (Asian-American), and genotype IV (Asian), all originating from South America. In order to address any potential uncertainties regarding the country of origin, we have included the metadata as supplementary material, which also contains the accession identifiers of the sequences utilized. It is important to note that initially, a total of 1305 sequences were retrieved. However, following the recommendations from IQTREE2, we proceeded with 1039 sequences for the subsequent phylogenetic analysis. We performed the following modifications in the “Materials and Methods” section.

Lines 154-158, Page 4: “…For the phylogenetic analysis, we retrieved a total of 1305 complete DENV genomes meeting the following criteria: belonging to DENV serotype 2, devoid of ambiguities, obtained from South America, and covering the same time period as our study. Our search yielded the genotypes I (American), genotype II (Cosmopolitan), genotype III (Asian-American), and genotype IV (Asian), all originating from South America. …”

The detection of Anelloviruses in most pools is interesting and may reflect the clinical status of these acutely symptomatic patients and recently described as a marker of immune incompetence. As a weakened immune system fails to control the virome normally residing in the body, these virus levels have been found to increase. Have the authors found this in other studies or could they relate the detection of TTV to a clinical relevance of the patients in the study?

Human anelloviruses, particularly Torque teno viruses (TTVs), have evolved a unique interaction with the host's immune system. Research has increasingly shown that the replication rate of these viruses can be a valuable biomarker for assessing the host's immune system functionality. In essence, the level of TTV replication may reflect the immunocompetence of an individual, serving as an indirect indicator of the host's immune system state. This relationship has practical implications, especially in the field of transplantation. For instance, in human solid organ transplantation, monitoring TTV levels could assist in the risk stratification of recipients, guiding immunosuppressive therapy by indicating whether the immune system is under- or overactive.

In Pool 15, we observed a significantly elevated number of Torque teno virus (TTV) reads, totaling 41,039. This high number might be likely attributable to the presence of an immunosuppressed patient within this pool. The predominant anelloviruses detected were classified within the Alphatorquevirus genus, specifically Torque teno virus strains 18, 22, and 24. However, the objective of our study was to identify viral agents related to hemorrhagic outcomes, instead of characterizing possible biomarkers especially in regards to human commensal viruses that can be used for evaluation of the host immunologic response.

However, Anelloviruses are circular DNA viruses. The authors need to discuss what reads were detected from TTV- how were they identified in a NGS library that amplified only polyA- RNAs. Did they detect TTV mRNA transcripts?

We are grateful for the valuable comment of the reviewer. We detected many different Anelloviruses including all three genera. This was reflected in the manuscript text under the following paragraph:

Lines 170-175, Page 5:  “… The virome of the tested patients was predominantly characterized by the presence of multiple commensal viruses, particularly from the Anelloviridae family: Genus Alpha-torquevirus (Torque teno virus – TTV), Betatorquevirus (Torque teno mini virus – TTMV) and Gammatorquevirus (Torque teno midi virus – TTMDV). Additionally, high numbers of reads for human pegivirus (HPgV-1) were identified in pools 7 and 18, which is also considered a commensal virus (Figure 2). …”

The extraction protocol was designed to isolate both viral RNA and DNA from the samples, including the genomes of the anelloviruses. Following the ligation of DNA fragments using ligase, these fragments underwent isothermal amplification, a process instrumental for the generation of genomic libraries. It is important to note that our library construction method does not selectively amplify only poly-A tailed RNAs, thereby allowing for a broader representation of the viral nucleic acid content in the generated libraries.

Table 1. Please reduce the number of decimal places.

We are grateful for the valuable feedback from the reviewer. To reduce the number of decimal places, we have chosen to display the first four columns in the thousand scale and have provided significant digits with three decimal places.

DISCUSSION:

The authors need to address why they failed to detect DENV-2 in the patient using the initial panel of diagnostic assays, both by their IgM serological assay as well as the molecular (RT-PCR?) assay. Did the new sequence contain SNPs at significant primer binding sites? Did the patient fail a serology control given their supposed immune suppression by being an oncology patient?  More explanation is needed.

As previously elucidated, testing for DENV RNA was not conducted at the Public Health Laboratory due to the following reasons: (1) The sample was collected six days after the onset of symptoms, whereas the established protocol mandates testing for DENV RNA by real-time PCR within five days of symptom onset. Furthermore, (2) the testing for DENV IgM yielded a negative result, suggesting that the sample was DENV-negative. However, it is worth noting that the patient's clinical condition, characterized by immune suppression due to oncologic disease, likely hindered the synthesis of antibodies. Therefore, the primary takeaway from this scenario is that the patient's clinical condition should always be taken into consideration, and molecular testing should be conducted regardless of the serological results.

The following explanation was added into the “Discussion” section:

Lines 228-230, Page 7 and Lines 231-238, Page 8: “... This case presented with diagnostic peculiarities, notably the negative DENV IgM serologic panel and sample collection occurring more than 6 days post-symptom onset, which initially precluded molecular confirmation. However, metagenomics subsequently revealed a high number of sequence reads, which were further confirmed by real-time PCR, showing a low cycle threshold (Ct=20). This low Ct value was indicative of acute DENV infection, suggesting that the patient's oncologic condition might have delayed or prevented the production of anti-DENV antibodies, leading to a false-negative initial serological result [33]. Therefore, the primary takeaway from this scenario is that the patient's clinical condition should always be taken into consideration, and molecular investigation should be conducted regardless of the serological results.…”

The library method chosen for the metagenomic study was designed for gene expression analysis of human polyA mRNA transcripts. This poses a great limitation since many pathogenic viruses do not encode a polyA mRNAs and would fail to be detected. The upstream use of DNAse is mentioned. What other endemic viruses were missed if they had chosen a different strategy?

We are appreciative of the valuable comment provided by the reviewer.

Initially, we treated the samples with DNase to eliminate host and bacterial DNA, following which viral nucleic acids were concentrated and extracted using a suitable kit designed for processing large volumes. It's important to note that this kit facilitates the extraction of both DNA and RNA. Subsequently, we conducted reverse-transcription and isothermal amplification using the Quantitect Whole Transcriptome Kit (QIAGEN), which is specifically designed for amplifying very low quantities of DNAs. It's worth mentioning that this kit does not exclusively amplify polyA transcripts. The preparation of libraries was based on the amplification of cDNA and the extracted DNA using a standard procedure based on the use of Illumina DNA prep. Based on our prior experience, we have found it to be highly suitable for the detection of various DNA and RNA viruses, including commensal viruses and those with pathogenic potential.

What do the authors plan to do with the large percentage of non-human, unmapped reads? What other approach could help identify the etiological agent of these patients?

In our study, we aimed to investigate the viruses associated with hemorrhagic fever. Hence, we utilized unmapped reads and classified them based on a viral database by Kraken2. However, similar to the approach used for a viral database, this method could also be applied to databases of other microorganisms such as bacteria. Unmapped reads serve as precursors for taxonomic classification and understanding of potential infections occurring in patients, which serve as potential new studies. Beyond unmapped reads, there are also unclassified reads, which are not assigned to any database. These reads, referred to as “dark matter”, underscore the limitations of our knowledge about the microorganisms in general. Within these unclassified reads, emerging viruses may be found, highlighting the need for extensive studies to identify them.

In a clinical diagnostic lab, what improvements are needed to detect highly pathogenic infectious agents in immune compromised patients? ...Especially if they could spread disease in a hospital setting.

This is an intriguing question that warrants a more detailed discussion.

In our view, given the immune suppression experienced by the patient, it is of paramount importance to introduce sensitive diagnostic tests capable of detecting very low viral loads. This may include the utilization of real-time PCR with high sensitivity or PCR tests designed to detect minimal quantities of nucleic acids, such as digital PCR. Additionally, next-generation sequencing and metagenomic analysis can play a role in confirming the infectious agent, as they offer sensitivity and the ability to detect virtually all genomic sequences, including those of viruses with significant clinical implications or unsuspected ones that could cause clinically relevant diseases in immunosuppressed patients. In this context, metagenomics can serve as an additional tool to achieve two main objectives: Firstly, to identify the agents responsible for clinical symptoms in this patient population, and secondly, to broaden the range of pathogens included in the detection panels to encompass underdiagnosed agents. Therefore, regular evaluation of viral pathogens among immunosuppressed patients is of paramount importance to understand their levels of circulation and clinical significance.

Furthermore, capacity building of human resources, particularly those responsible for laboratory techniques and diagnosis, along with interdisciplinary integration of clinical and laboratory professionals, is crucial for the evaluation and appropriate diagnosis of emerging viruses in immunosuppressed patients.

The following explanation was added in the “Discussion” section:

Lines 239-254, Page 8:  “... The diagnosis of viral infections in patients with immunosuppression poses a formidable challenge, primarily due to the compromised serological responses often observed in this demographic. Consequently, it becomes critically important to integrate diagnostic methodologies endowed with the sensitivity to identify low viral loads, such as real-time PCR or digital PCR. Further augmenting the diagnostic arsenal, next-generation sequencing and metagenomic analysis offer unparalleled sensitivity and the comprehensive capability to detect a vast array of genomic sequences. This includes the genomes of viruses that carry significant clinical implications, thereby facilitating the accurate identification of the infectious agents involved. Moreover, the enhancement of human resource capabilities, particularly those personnel engaged in the execution of laboratory techniques and diagnostics, emerges as a pivotal factor. This enhancement, coupled with the interdisciplinary integration of clinical and laboratory professionals, is essential for the efficacious evaluation and accurate diagnosis of emergent viral pathogens in immunosuppressed populations. Such an integrated approach not only fosters precision in diagnostics but also significantly contributes to the broader objective of advancing patient care and management in the face of evolving viral threats.…”

Overall, this is an unsurprising result that needs a few more details for completeness. Often metagenomics studies fail to detect a pathogenic agent, but the data needs to be submitted to public, accessible databases to allow for reanalysis by different bioinformatic tools.

We fully recognize the imperative of data sharing within the scientific milieu to foster collaborative research and expedite advancements in our field. However, after a thorough deliberation process, taking into account a multitude of factors and principally the integrity of ongoing and forthcoming research (master thesis), we reached the decision to refrain from making the sequencing reads publicly available at this juncture. This omits the complete DENV-2 genome, which, following the reviewer's recommendation, has been deposited in the GISAID and NCBI databases. We are open to discussions on our research findings, potential collaborative ventures, and, where feasible, providing access to our data under specific agreements that safeguard the privacy and ethical considerations pertinent to this work.

Line 117-118. Replace molecular serotyping with genotyping.

The molecular test we employed was for the differentiation of DENV serotypes through amplification. Consequently, we referred to the process as "molecular serotyping." However, following the reviewer's suggestion, we have replaced "molecular serotyping" with "molecular genotyping."

Line 87: typo: into

Error corrected.

Reviewer 2 Report

Comments and Suggestions for Authors

            The authors evaluated the presence of pathogens in cases diagnosed with hemorrhagic fever in Minas Gerais. Brazil, with no etiological agent detect by the classical testing panel. NGS of pooled sample allowed to detect dengue virus in 1/43 samples. Although the information is important and interesting, many concerns limit the relevance of this study and should be addressed before publication.

1.       Title: the authors should not use the term viral, since the etiological agent is unknown. Indeed, their testing panel include Rickettsia and Leptospira.

2.       Abstract. The mention of pooling of samples for NGS is confusing. The samples are pooled in groups of 2-3, but for then a different index was used, I supposed. Thus information is available for each group and not really pooled.

3.       Abstract. There is too much information on the advantages of NGS and few information on the results of this study.

4.       Introduction, first sentence. The authors mention the most important viral etiological agents of hemorrhagic fever, and include as expected the arenaviruses. However, arenaviruses were not included in the testing panel analyze before NGS. Instead, hantavirus, which is not so frequently associated with hemorrhagic fever in the region, is included in the test. This should be discussed.

5.       Material and Methods: The authors mentioned that they used Quan-94 tiTect Whole Transcriptome Kit, which basically produce a cDNA from the RNAs present in the sample. They should describe a little bit more their protocol.

6.       Results line 160: an extraordinary number of… reads? in pool 15.

7.       One complete genome was assembled from one sample, consisting of dengue virus 2 Cosmopolitan genotype. However, the reads came from a pool of samples. Even if only one was positive for dengue when split, the other samples may have dengue virus RNA in low proportion that could not be detected in the real time. In fact, non of these samples were detected previously by the testing panel applied to them. This limitation should be discussed.

8.       Discussion line 196-197: presented with

9.       Discussion line 201: the oncologic condition of the patient may have reduced the possibility of antibody detection, but the authors mentioned that they performed PCR in the samples. Why this sample did not arise a positive PCR reaction at the beginning?

10.    In conclusion, the experimental strategy proposed (NGS of pool of 2-3 samples) did not lead to an increased sensitivity in detecting the etiological agent of hemorrhagic fever. Only for one case of 43, dengue virus was detected. This might also be due to the NGS performed on cDNA or to the sample pooling. This should be discussed.

11.    The conclusion of lines 226 is then not completely right.

12.    The limitations of this study should be discussed. In addition, a positive control of pooled samples with 3 etiological agents (two viruses and one leptospira, for example) should have been advisable to validate the sensitivity of the proposed pooled strategy.

Comments on the Quality of English Language

Some minor editing, decsribed in the comments.

Author Response

REVIEWER#2

The authors evaluated the presence of pathogens in cases diagnosed with hemorrhagic fever in Minas Gerais. Brazil, with no etiological agent detect by the classical testing panel. NGS of pooled sample allowed to detect dengue virus in 1/43 samples. Although the information is important and interesting, many concerns limit the relevance of this study and should be addressed before publication.

We are profoundly thankful for the positive feedback provided by the reviewer regarding our study. The provided comments are greatly appreciated. Thank you very much!

  1. Title: the authors should not use the term viral, since the etiological agent is unknown. Indeed, their testing panel include Rickettsia and Leptospira.

We sincerely appreciate the insightful feedback provided by the reviewer. The reviewer's observation regarding the various pathogenic agents capable of causing hemorrhagic fevers, including bacteria, particularly Rickettsia and Leptospira, is indeed valid. As a result, we have modified the title accordingly by removing the term "viral" and enhancing it as follows:

New title: “METAGENOMIC ANALYSIS FOR DIAGNOSIS OF HEMORRHAGIC FEVERS IN MINAS GERAIS, BRAZIL”

  1. Abstract. The mention of pooling of samples for NGS is confusing. The samples are pooled in groups of 2-3, but for then a different index was used, I supposed. Thus information is available for each group and not really pooled.

We totally reformulated our Abstract. Now we added the number of prepared pools and extended the results section.

Lines 31-33, Page 1: “... The samples were grouped into 18 pools according to geographic origin and analyzed through next-generation sequencing on the NextSeq 2000 platform. …”

  1. Abstract. There is too much information on the advantages of NGS and few information on the results of this study.

We are grateful for the valuable comment of the reviewer. We extended the presentation of the results in the “Abstract” section:

Lines 35-41, Page 1:  “… Further verification via real-time PCR confirmed the presence of DENV-2 RNA in an index case involving an oncology patient with hemorrhagic fever, who had initially tested negative for anti-DENV IgM antibodies, thereby excluding this sample from initial molecular testing. The complete DENV-2 genome isolated from this patient was taxonomically classified within the cosmopolitan genotype that was recently introduced into Brazil. These findings highlight the critical role of considering the patient's clinical condition when deciding upon the most appropriate testing procedures. …”

  1. Introduction, first sentence. The authors mention the most important viral etiological agents of hemorrhagic fever, and include as expected the arenaviruses. However, arenaviruses were not included in the testing panel analyze before NGS. Instead, hantavirus, which is not so frequently associated with hemorrhagic fever in the region, is included in the test. This should be discussed.

We appreciate the valuable feedback provided by the reviewer. It has been noted that the routinely used diagnostic panel did not include testing for arenavirus. Despite the severe clinical manifestations associated with Brazilian Hemorrhagic fever, caused by the mammarenavirus Sabia, there have been only four documented cases in Brazil, primarily in the State of São Paulo. In the State of Minas Gerais, hemorrhagic fevers caused by arenaviruses have never been documented. In cases of suspicion of such an infection, samples are typically tested in a national reference laboratory rather than at a state level. Due to these factors, including cost and the sporadic nature of Brazilian hemorrhagic fever cases, the inclusion of arenavirus testing in the diagnostic panel was not deemed feasible. On the other hand, the occurrence of Hantavirus is more prevalent in the Minas Gerais state. Between 2012 and 2023, there were 78 confirmed cases of the disease with a lethality rate of 53%.

  1. Material and Methods: The authors mentioned that they used Quan-94 tiTect Whole Transcriptome Kit, which basically produce a cDNA from the RNAs present in the sample. They should describe a little bit more their protocol.

We are grateful for the questioning of the reviewer. In fact, we first synthesize the cDNA from the extracted viral RNA using the Superscript III First-Strand Synthesis System. Then the cDNA (and the extracted DNA) is submitted to isothermal amplification using the Quantitect Whole Transcriptome Kit. The kit itself is used for amplification of the cDNA rather than reverse transcription. The amplified by isothermal amplification DNA is subsequently submitted for genomic library preparation. We apologize that it has not been clear for the reviewer the library preparation protocol and for that reason we revised the text of the “Materials and Methods” in introduce more clear explanation of this protocol:

Lines 115-118, Page 3: “... Reverse transcription was conducted utilizing the Superscript III First-Strand Synthesis System (ThermoFisher Scientific). The amplification of the cDNA was performed employing the QuantiTect Whole Transcriptome Kit (QIAGEN), following an isothermal strategy. …”

  1. Results line 160: an extraordinary number of… reads? in pool 15.

We modified this phrase like this “...Among the viruses of clinical importance, DENV was identified with a high number of sequence reads in pool 15 (15,791,645 reads). …”

We changed this phrase by adding the following sentence in the revised version of the manuscript.

Lines 184-185, Page 6: “… a high number of sequence reads in pool…”

  1. One complete genome was assembled from one sample, consisting of dengue virus 2 Cosmopolitan genotype. However, the reads came from a pool of samples. Even if only one was positive for dengue when split, the other samples may have dengue virus RNA in low proportion that could not be detected in the real time. In fact, non of these samples were detected previously by the testing panel applied to them. This limitation should be discussed.

We appreciate the valuable comment provided by the reviewer, and we would like to offer the following clarification. All the samples comprising the pools were tested using a highly sensitive PCR method capable of detecting all DENV serotypes, as detailed in the "Materials and Methods" section. Furthermore, we utilized a specific panel of primers and probes for DENV serotype differentiation, which yielded positive results only for the index sample. Based on this, we believe that the likelihood of very low DENV loads in the other samples within this pool (3 samples) is exceedingly low.

It is important to note that the sample in question was not classified as negative by the locally applied diagnostic kit for DENV PCR detection. Rather, it was not tested for DENV RNA via PCR in the Public Health Laboratory due to its negative DENV-IgM status. Consequently, it was considered negative for DENV and subsequently selected for metagenomic analysis to elucidate the etiological cause of the hemorrhagic disease.

  1. Discussion line 196-197: presented with

We performed this correction.

  1. Discussion line 201: the oncologic condition of the patient may have reduced the possibility of antibody detection, but the authors mentioned that they performed PCR in the samples. Why this sample did not arise a positive PCR reaction at the beginning?

This is an important issue raised by the reviewer. The patient in question presented with hemorrhagic disease, but due to the negative result obtained for the DENV-IgM antibodies, it was considered negative for DENV, and confirmatory PCR for DENV was not performed. Additionally, the clinical sample was collected six days after the onset of the symptoms, and the Public Health Laboratory protocol dictates five days as a maximum period for DENV molecular testing. Therefore, it is of paramount importance to consider the clinical condition of the patient when determining diagnostic algorithms and conducting more detailed investigations. Collaboration between laboratory personnel and clinicians is crucial in facilitating an open conversation to establish the best diagnostic algorithm for each particular case.

We added the following explication in the discussion section:

Lines 239-254, Page 8: “... The diagnosis of viral infections in patients with immunosuppression poses a formidable challenge, primarily due to the compromised serological responses often observed in this demographic. Consequently, it becomes critically important to integrate diagnostic methodologies endowed with the sensitivity to identify low viral loads, such as real-time PCR or digital PCR. Further augmenting the diagnostic arsenal, next-generation sequencing and metagenomic analysis offer unparalleled sensitivity and the comprehensive capability to detect a vast array of genomic sequences. This includes the genomes of viruses that carry significant clinical implications, thereby facilitating the accurate identification of the infectious agents involved. Moreover, the enhancement of human resource capabilities, particularly those personnel engaged in the execution of laboratory techniques and diagnostics, emerges as a pivotal factor. This enhancement, coupled with the interdisciplinary integration of clinical and laboratory professionals, is essential for the efficacious evaluation and accurate diagnosis of emergent viral pathogens in immunosuppressed populations. Such an integrated ap-proach not only fosters precision in diagnostics but also significantly contributes to the broader objective of advancing patient care and management in the face of evolving viral threats.…”

  1. In conclusion, the experimental strategy proposed (NGS of pool of 2-3 samples) did not lead to an increased sensitivity in detecting the etiological agent of hemorrhagic fever. Only for one case of 43, dengue virus was detected. This might also be due to the NGS performed on cDNA or to the sample pooling. This should be discussed.

The pooling of samples was conducted with the objective of reducing the sequencing costs rather than primarily aiming to enhance sensitivity. Simultaneously, we opted for a reduced number of samples per pool to elevate the sequencing depth. This strategy was evident in the high number of DENV sequence reads obtained, including the increased depth of the complete genome. The detection of etiological agents through next-generation sequencing relies on their presence in the sample. Therefore, we employed viral metagenomics to investigate at what level the conventional virological diagnosis may fall short in identifying these viruses.

  1. The conclusion of lines 226 is then not completely right.

We performed different modifications in the conclusion section, softening our vision on the obtained results and the clinical use of metagenomics. The conclusion was modified by the following manner.

Lines 285-290, Page 9:  “... In conclusion, viral metagenomics has demonstrated its value in identifying the viral origin of hemorrhagic fevers in the State of Minas Gerais, Brazil, as evidenced by the confirmation of a DENV-2 cosmopolitan genotype in a patient with negative DENV IgM serology. Additionally, viral metagenomics provides deeper insights into the etiology of diseases that cannot be attributed to routine diagnostic tests alone. However, additional confirmatory tests, mostly direct molecular methods are recommended to ensure accuracy and reliability. …”

Moving forward, further research is necessary to effectively incorporate viral metagenomics into routine clinical diagnostics. This includes refining protocols, optimizing bioinformatics pipelines, and establishing standardized guidelines for result interpretation. By doing so, we can harness the full potential of viral metagenomics to enhance our understanding of infectious diseases and improve patient care outcomes.

  1. The limitations of this study should be discussed. In addition, a positive control of pooled samples with 3 etiological agents (two viruses and one leptospira, for example) should have been advisable to validate the sensitivity of the proposed pooled strategy.

As with all studies, ours also presents certain limitations. A primary limitation is the focus solely on viral pathogens with steps primarily geared towards depletion of bacterial DNA. Hemorrhagic infections can also be caused by various bacteria. These include but are not limited to rickettsial infections (Brazilian spotted fever), shigellosis, hemorrhagic salmonellosis, meningococcemia, leptospirosis that all have very challenging diagnosis. However, specific confirmation of these infections is needed but this was not an objective of our study. Additionally, our investigation was constrained by the utilization of an established Reference Sequence Database containing known viral genomes. Consequently, unclassified reads were not thoroughly examined, potentially resulting in the oversight of emerging viruses possibly associated with hemorrhagic diathesis. These limitations underscore the need for comprehensive approaches in future studies, which may involve expanding the scope to include bacterial pathogens and refining methodologies to capture emerging viruses. Such efforts are essential for advancing our understanding of the complex etiology of hemorrhagic fevers and improving diagnostic capabilities.

It's worth noting that we previously evaluated the sensitivity of our protocol for viral identification in pooled samples and found it to be suitable and adequately sensitive. For further details on this aspect, please refer to (https://doi.org/10.1016/j.tracli.2020.07.001).

The following paragraph was added into the revised version of the manuscript:

Lines 266-281, Page 8 and Lines 282-283, Page 9:   “... As with all studies, ours also presents certain limitations. A primary limitation is the focus solely on viral pathogens, with steps primarily geared towards depletion of bacterial DNA. Hemorrhagic diseases can also be caused by various bacteria, including but not limited to rickettsial infections (such as Brazilian spotted fever), shigellosis, hemorrhagic salmonellosis, meningococcemia, and leptospirosis, all of which pose significant challenges in diagnosis. A more meticulous search in the bioinformatic data obtained for bacteria revealed abundant reads of Leptospira interrogans in pool 18, and pools 8 and 13 showed high sequence read numbers for Rickettsia rickettsii, which causes Brazilian spotted fever, a hemorrhagic disease associated with high mortality rates [39]. However, specific confirmation of these infections was beyond the scope of our study. Additionally, our investigation was constrained by the utilization of an established Reference Sequence Database containing known viral genomes. Consequently, unclassified reads were not thoroughly examined, potentially resulting in the oversight of emerging viruses possibly associated with hemorrhagic diathesis. These limitations underscore the need for comprehensive approaches in future studies, which may involve expanding the scope to include bacterial pathogens and refining methodologies to capture emerging viruses. Such efforts are essential for advancing our understanding of the complex etiology of hemorrhagic fevers and improving di-agnostic capabilities. …”

Round 2

Reviewer 2 Report

Comments and Suggestions for Authors

The authors addressed satisfactorely the concerns of the original version.